# Development and validation of a casemix classification to predict costs of specialist palliative care provision across inpatient hospice, hospital and community settings in the UK: a study protocol

Ping Guo,[1] Mendwas Dzingina,[1] Alice M Firth,[1] Joanna M Davies,[1] Abdel Douiri,[2] Suzanne M O'Brien,[1] Cathryn Pinto,[1] Sophie Pask,[1] Irene J Higginson,[1] Kathy Eagar,[3] Fliss E M Murtagh[4]

For numbered affiliations see end of article.

**Correspondence to**
Dr Ping Guo;
ping.guo@kcl.ac.uk

## ABSTRACT

**Introduction** Provision of palliative care is inequitable with wide variations across conditions and settings in the UK. Lack of a standard way to classify by case complexity is one of the principle obstacles to addressing this. We aim to develop and validate a casemix classification to support the prediction of costs of specialist palliative care provision.

**Methods and analysis** Phase I: A cohort study to determine the variables and potential classes to be included in a casemix classification. Data are collected from clinicians in palliative care services across inpatient hospice, hospital and community settings on: patient demographics, potential complexity/casemix criteria and patient-level resource use. Cost predictors are derived using multivariate regression and then incorporated into a classification using classification and regression trees. Internal validation will be conducted by bootstrapping to quantify any optimism in the predictive performance (calibration and discrimination) of the developed classification. Phase II: A mixed-methods cohort study across settings for external validation of the classification developed in phase I. Patient and family caregiver data will be collected longitudinally on demographics, potential complexity/casemix criteria and patient-level resource use. This will be triangulated with data collected from clinicians on potential complexity/casemix criteria and patient-level resource use, and with qualitative interviews with patients and caregivers about care provision across difference settings. The classification will be refined on the basis of its performance in the validation data set.

**Ethics and dissemination** The study has been approved by the National Health Service Health Research Authority Research Ethics Committee. The results are expected to be disseminated in 2018 through papers for publication in major palliative care journals; policy briefs for clinicians, commissioning leads and policy makers; and lay summaries for patients and public.

**Trial registration number** ISRCTN90752212.

### Strengths and limitations of this study

► This is the first study in the UK to determine the variables and potential classes to be included in a casemix classification, which predicts costs of specialist palliative care provision across inpatient hospice, hospital and community settings.

► Transparent Reporting of a multivariable prediction model for Individual Prognosis Or Diagnosis is used to guide the reporting of this study.

► The newly developed casemix classification in phase I will be externally validated by using a different data set in phase II.

► This study also promotes the implementation of outcome measures in palliative care into routine clinical practice across different settings of care.

► This is a UK-focused study so the casemix classification may not be directly applied to palliative care in other countries without further investigation and refinement.

## INTRODUCTION

People with advanced and incurable illness often suffer complex and multiple symptoms and psychosocial concerns because of their illness or impending death.[1 2] Their families may provide day-to-day care, as well as be affected by their own anxieties, concerns and potential losses. These bring increased need for health and social care, with need being defined as 'the ability to benefit from health or social care interventions'.[3] Palliative care has developed to meet the needs of these patients and families, which addresses physical/psychological symptoms and provides social, practical and spiritual support. The UK ranks first in the 2015 Quality of Death

Index—a measure of the quality of dying in 80 countries,[4] and the hospice movement in the UK has provided a model of good palliative care for those in need.

However, marked inequities exist in provision of palliative care across England. Older people and those with non-cancer diagnoses are less likely to access specialist palliative care.[5 6] There are also major geographical variations, ranging from £186 to £6213 per person across different primary care trusts in 2010,[7 8] often resulting in a poor match between individual needs, resources provided to meet those needs and health outcomes achieved. With an ageing population and increasing rates of chronic diseases, the growing healthcare burden is overwhelmingly challenging in terms of health resource allocation around the UK.[9] With recognition of the constraints on resources, there is support for the systematic approach to mapping individual needs accurately and improving the quality and efficiency of palliative care. This has been endorsed as a high priority nationally.[7]

The Diagnosis-Related Group is a useful classification of healthcare needs driven by the diagnosis but is inappropriate for palliative care, because palliative care needs are not driven by the diagnosis but by factors such as functional status and symptoms. Palliative care needs a consistent method of classifying types of patients with complexity of needs, treatment and costs, using casemix criteria.[10–12] It is necessary to identify those with more complex palliative needs, requiring more resources. An Australian casemix classification for palliative care was developed in 1997, empirically tested and progressively refined over time.[13–15] The Australian casemix classification consists of classes defined by five criteria including phase of illness, problem severity, functional status and dependency, age and model of care, as most strongly predictive of resource use.[16] Full class definition and categorisation are available at http://www.pcoc.org.au/. Its implementation proved the possibility of consistently and routinely collecting data in practice in Australia.[16] However, due to variations in outcome measures collected and palliative care provided between countries, it is unclear whether any existing palliative care classification can be easily applied to the UK to address unmet needs and resolve the inequity.

This study is part of a 5-year National Institute for Health Research-funded C-CHANGE programme (RP-PG-1210–12015). It aims to develop and validate a casemix classification for palliative care in the UK. Specific objectives are:

► To determine the cost predictors of specialist palliative care, adjusting for important confounding factors including unmet needs.
► To develop and validate a casemix classification for specialist palliative care.

## METHODS AND ANALYSIS

The Transparent Reporting of a multivariable prediction model for Individual Prognosis Or Diagnosis statement[17] is used to guide the reporting of this study protocol.

### Study design and source of data

This study consists of two phases: development phase (phase I) and validation phase (phase II). In the analysis part of phase I, we will identify variables that predict costs to be included in casemix classification through a cohort study with only clinician data, and develop potential casemix classes for palliative care which will be tested in individual episodes of care (internal validation). In phase II, we will prospectively validate these potential classes in a new cohort study with patient, caregiver and clinician data, and include qualitative interviews to ensure it works during care transitions and longitudinally over the course of illness (external validation).

### Phase I: development phase

This is a cohort study to determine the variables and potential classes to be included in a casemix classification, based on individual episodes of care across inpatient (hospice and hospital) and community settings. An 'episode of care' starts when a patient is admitted to inpatient services or begins to receive specialist palliative care from a community-based or outpatient service, and ends when a patient is discharged from that service or dies. The median duration of an episode of care is expected to be under 14 days in inpatient and at a median of 72 days in community settings.[18]

Phase I was conducted between 31 July 2015 and 30 September 2016 and follow-up ended on 30 November 2016. Data were collected from clinicians through surveys, including demographic/clinical data, episode start/end data, patient attributes that predict palliative care resource consumption (eg, phase of illness, functional status and problem severity as identified in the Australian casemix classification study[13 14 16]), plus information on patient-level resource use (staff activity and clinical services) in specialist palliative care settings.

### Phase II: validation phase

This phase is a mixed-methods cohort study with a concurrent nested design:[19]

► Quantitative main component—prospectively collect data from patients, caregivers and clinicians on palliative care needs, concerns, outcomes and resource use.
► Qualitative nested component—longitudinal interviews with a subsample of participants to understand care provision in each setting and transitions between settings.

Phase II will be conducted between 1 November 2016 and 30 April 2018, with follow-up ending on 31 May 2018. The same variables and measurements as phase I will be collected from clinicians. Additionally, patient/caregiver participants will provide data on symptoms/concerns, experience of care and their use of services, and will be followed through all episodes of care from recruitment to death or the end of this study. A post bereavement survey will be conducted with caregivers where appropriate to identify symptoms/concerns immediately prior to death and support needs after death.

## Participants

### Phase I: development phase

All adults (≥18 years) receiving specialist palliative care newly admitted in 10 participating sites during the study period (two sites providing hospital advisory services, one providing community-based service only, six providing hospice inpatient and community-based services, one providing hospital advisory and community-based services) were included, regardless of primary diagnosis. We selected these sites with the aim of ensuring a representative sample in terms of population demographics (age distribution, ethnicity, socioeconomic status and rural/urban composition). Written informed consent was taken by a qualified member of the research or clinical team. This phase included individuals with limited, fluctuating, diminishing or lack of capacity, vital to ensure that any casemix classification is applicable to all palliative patients, not just those able to consent. It is recognised that a high proportion of those have impaired capacity, and these patients may need palliative care and resources most.[20–23] Therefore, if the clinician assessed that the patient did not currently have the capacity to give consent, assent was sought from an accompanying family member, or failing that, a staff member to whom the patient was known. Where a formally appointed power of attorney existed, this took precedence.

### Phase II: validation phase

All adults (≥18 years) receiving specialist palliative care newly admitted from 14 sites during the study period (participating sites in phase I, but extending to four additional sites to increase recruitment) will be eligible, regardless of primary diagnosis. Additional sites were accepted if they could contribute to maintaining or extending representativeness in terms of population demographics (age distribution, ethnicity, socioeconomic status and rural/urban composition). Written informed consent will be sought from eligible patients (and their family caregivers) at the outset, including advance consent for study follow-up if capacity is lost. This study is of minimal risk with participants not exposed to undue harm.[24]

For both phase I and II, a list of study sites can be obtained via https://www.kcl.ac.uk/nursing/departments/cicelysaunders/research/studies/c-change/c-change.aspx.

Since transition of care is important for casemix, patient participants who transfer between settings (and caregiver participants) will be followed up. We will verbally confirm continuing consent with each participant when making such contact. If capacity of patient participants is lost, we will seek advice from a personal consultee on whether the patient should remain in the study to verify no change of decisions has been expressed by the patient prior to loss of capacity. On recruitment, participants will be asked if they are willing to be interviewed at a later stage. To capture variation in age, gender, diagnosis and geographical location, we will purposively select 20–25 patients and family caregivers with at least two transitions of care for face-to-face interviews. Each semistructured interview will last 40 min, but will be guided by each participant. In order to provide information on how care transitions might be better negotiated to improve outcomes and experiences, the interviews will cover communication, coordination of care, information/support needs, discharge planning and experience of transitions.

## Outcome

The outcome is cost of specialist palliative care (including per diem cost, per phase cost, and total episode cost) captured by: (1) staff activity matrix in both the development and validation phase, (2) the Palliative care Resource Use Score (PRUS) in both phases and (3) Palliative Client Services Receipt Inventory (Pall-CSRI) in validation phase only.

### Staff activity

At every contact, nurses, doctors and allied health professionals will record the time spent on face-to-face and phone contacts, and patient-level administrative time per shift using the staff activity matrix paper version (table 1). Staff training and site feedback of activity data were conducted regularly to improve and optimise data quality.

### Palliative care Resource Use Score

PRUS is a questionnaire specifically designed to capture palliative care resource use in a standardised way. It will be collected by the staff at change of 'phase of illness'[25 26] and at the end of each episode by recording the following information for the phase which has ended:

► Level of professional input (eg, registered nurses, specialist palliative nursing staff, palliative doctors and social workers) and whether this met patient/family needs.
► Level of 'out-of-hours' services by professional designation.
► Equipment, high-cost drugs, diagnostic tests and medical imaging. Within PRUS, the member of staff is asked to indicate whether or not equipment, high-cost oral/transdermal medications, injectable medications/interventions and medical imaging or tests were received by the participant, from a list derived from prior work to identify these options.

| Table 1 | Staff activity matrix (each box is completed in units of 5 min, from 0 up to 120 min) | | | |
|---|---|---|---|---|
| Staff time (mins) | Patient | Family/carer | Professional (internal) | Professional (external) |
| Face-to-face/phone time | | | | |
| Administrative time | | | | |

► Any unmet needs which provide insight into any gaps between needs and provision, as identified by a member of staff.[27–29] When completing PRUS, the member of staff is asked to estimate retrospectively if there were any unmet needs (yes/no) in the care provided by healthcare assistants, registered nurses, palliative medicine doctors and allied healthcare professionals. This member of the staff is also asked whether there were any unmet needs in out-of-hours care, and whether the equipment provided met the patient's needs. If there were unmet needs identified in any of these areas above, a free text field is provided to specify these unmet needs.

## Palliative Client Services Receipt Inventory

The Pall-CSRI is a patient/caregiver completed inventory of palliative care services received, and adapted from those used with palliative care populations.[30] It takes approximately 20 min to complete and collect retrospective information about the use of health/social care services, medication, living situation, income, employment and benefits, plus informal care. The Pall-CSRI will be collected once every three months or at the end of each episode of care, whichever is earlier in the validation phase.

In the development phase, only direct care costs will be calculated but not productivity losses.[31] Direct care cost refers to all costs due to resource use that are completely attributable to the use of a healthcare intervention or illness, which can be split into direct medical costs (the cost of a defined intervention and all follow-up costs for other medication and healthcare interventions in ambulatory, inpatient, nursing care, home or other relevant settings) and direct non-medical costs (eg, transportation costs and additional paid caregiver time).[32] In the validation phase, we will use the same health services costing perspective as in the development phase, but informal care costs are also considered to be in-scope and will be analysed separately, hypothesising that informal care costs are (1) greater in those with unmet needs, and (2) greater in those with non-cancer conditions. The costs of each resource item will be calculated by combining the resource use data with appropriate unit cost obtained from recognised sources including the annual compendium produced by the Personal Social Services Research Unit,[33] prices on the National Health Service (NHS) supply chain website http://www.supplychain.nhs.uk, British national formulary[34] and NHS reference costs.[35] The focus will be on staff time—the main resource in palliative care documented in a variety of settings and countries.[36–38] We will attribute costs according to a standard costing methodology adopted from the current NHS costing principles.[39]

## Predictors (casemix criteria)

Proposed cost predictors can be categorised into three groups: (1) predictors collected from clinicians, (2) predictors collected from patients/family caregivers and (3) model of care.

**Table 2** Data from clinicians in both development phase and validation phase

| Type of data | Proposed cost predictors |
| --- | --- |
| Demographic data | Age |
| | Gender |
| | Postcode |
| | Ethnicity |
| | Marital status |
| | Living circumstances |
| | Need for interpreter |
| | Setting of care |
| Clinical data | Primary diagnosis |
| | Secondary diagnoses |
| | Comorbidities |
| Episode start and end data | Episode start date |
| | Episode end date |
| | Endpoint of episode (discharged or died). |
| | Discharge destination, if discharged |
| Key casemix | Phase of illness at start of episode |
| | Functional status (AKPS) |
| | Dependency (Modified Barthel Index) |
| | Problem severity (IPOS staff version) |
| | Family/caregiver needs |

AKPS, Australia-modified Karnofsky Performance Status; IPOS, Integrated Palliative care Outcome Scale.

### Predictors collected from clinicians

In both the development and validation phases, clinicians will record demographic, clinical and episode start administrative data. Episode end data will be collected at the end of each episode of care. Casemix data include phase of illness (stable, unstable, deteriorating, dying), functional status (measured by Australia-modified Karnofsky Performance Status (AKPS)), dependency (Modified Barthel Index), problem severity (Integrated Palliative care Outcome Scale—IPOS staff version) and caregiver needs (table 2). The measures are available directly from phase of illness (www.pcoc.org.au), AKPS,[40] Modified Barthel Index[41] and IPOS (www.pos-pal.org), respectively. Phase of illness will be assessed daily for people receiving inpatient care and at each face-to-face contact basis for those receiving community-based care. Other variables including AKPS, Modified Barthel Index, IPOS staff (and caregiver needs when feasible) will be collected at the start of episode, end of episode and at change of phase of illness.

### Predictors collected from patients and family caregivers

Variables (table 3) will be collected in the validation phase from patients/caregivers using face-to-face/

**Table 3** Predictors collected from patients and family caregivers in validation phase only

| Data collection from patients | Data collection from family caregivers |
| --- | --- |
| IPOS patient version | Basic demographic information |
| Distress Thermometer | Distress Thermometer |
| Views on Care | Two caregiver questions |
| SF-12v2* | Zarit six items |
| Patient experiences of integrated care | |

*The measure could be completed by a family caregiver if the patient is too unwell to complete.
SF -12v2, Short-Form Health Survey V.2.0.

telephone contacts and postal questionnaires according to their preferences. IPOS patient version and Distress Thermometer will be collected from patients at the start of episode, at change of phase of illness and at the end of episode. Views on Care and Short-Form Health Survey V.2.0 (SF-12v2) will be collected at the start and end of each episode of care. Selected questions capturing experiences of integrated care[42] will be collected at the end of episode only.

All caregiver-reported measures will be collected at the start of episode, at change of phase of illness and at the end of episode. Where phase length exceeds 4 weeks, data collected from patients (and caregivers) will be captured as a change of phase. If the patient is transferred to an out-of-scope inpatient setting, we will follow up with the patient when he or she gets home and retrospectively collect minimum information about this episode of care.

If the patient is deceased, data such as SF-12v2 and Pall-CSRI will be collected from the participating caregiver three months after death by postal questionnaire, along with the bereavement support information and contacts. The time between death and the postal questionnaire may influence caregivers' willingness to share their views; shorter periods possibly being too upsetting to contemplate involvement.[43] There is mixed evidence of the best time to make contact with potential participants for follow-back surveys[44 45] and 3–4 months post bereavement is considered acceptable by bereaved families.[30]

## Model of care
Models of care may be a stronger cost driver than patient-level variables.[13 14 16] Specialist palliative care services have various configurations of staff, interventions and other characteristics. Currently, there is no consistent way to define models of palliative care. A separate, parallel study will be conducted in which service-level data including the numbers, disciplines and grading of staff, the nature/duration of their involvement and use of volunteers will be collected to comprehensively characterise different models of palliative care provision in the participating sites. With findings from this step, a new categorical

variable will be created representing 'model of care' to be included in our analysis.

## Sample size
In the development phase, based on standard recommendations for fitting multivariate models, a minimum of 50+8×m (where m is the number of predictors) is required to test the hypothesis that the population multiple correlation equals zero with a power of 80%, alpha=5% and a medium effect size for the regression analysis ($R^2$=0.13).[46] We estimated a sample size of 450 episodes per setting (allowing for 25% incomplete episodes and up to 10% of complete episodes being cost outliers with unusually high or low costs). A total of 1350 patient episodes across settings is needed.

In the validation phase, again a minimum of 50+8×m (where m is the number of predictors) is required to test the hypothesis that the population multiple correlation equals zero with a power of 80%, alpha=5% and a medium effect size for the regression analysis ($R^2$=0.13).[46] In phase II, the number of predictors reflects the casemix variables, and needs to be clinically relevant[47] to ensure a meaningful casemix classification. We expect the final casemix classification to contain 5–8 casemix variables, based on the Australian experience.[13] We have estimated that data from 114 (50+(8×8)) episodes of care per setting (hospice, hospital, community) are needed. Assuming an average of three episodes per participant[15] in this longitudinal phase, this represents at least 38 participants per setting. Allowing for 50% attrition, we estimate needing a total of 228 participants, which will provide about 684 patient episodes. The sample size will be recalculated and inflated using the design effect based on intraclass correlation which will be estimated through interim analyses. A total of 300 participants across settings is needed, including extra 25% participants allowing for intraclass correlation.

## Missing data and data handling
Data will be collected prospectively and recorded on an electronic database. Any data transferred from the sites to the central study database will be carried out under the NHS Code of Practice on Confidentiality. Data will be cleaned and cross-checked using a number of internal checks (automated data validation on entry, independent cross-checking of a 5% sample of data from each site and range checking across all data types) to track errors and inconsistencies and amend where possible. Checked data will be transferred to statistical software (Stata SE V.12) for analysis. Missing value analysis will be used to quantify missing values and understand the reason for missing data: dropout of participants, errors in data entry and missing with no identifiable reasons. We will adopt multiple imputation technique based on Markov Chain Monte Carlo techniques if the percentage of observed missing values in independent variables exceeds 10% provided values are missing at random, and carry out sensitivity analysis for effects on casemix

classification, predictive validity and misclassification, where feasible.

## Statistical analysis methods

We will use descriptive statistics including frequencies, mean and SD, and median and interquartile ranges as appropriate to describe the sample characteristics and number and length of episodes of care. Trajectory of outcomes and costs will be described, comparing patterns in patients with cancer and without cancer, and contrasting those with no unmet needs and those with unmet needs. We will compare our data with existing phase and episode data from the Australian Palliative Care Outcome Collaboration.[48] In the Australian classification, five casemix criteria (phase of illness, problem severity, functional status and dependency, age, model of care) were found to be most strongly predictive of resource use. If our analyses demonstrate the same five criteria as most predictive of resource use, the Australian classification or a refinement of it will be adopted. If there is disparity, then the data on individual casemix criteria will be used to develop a new casemix classification, using a recursive partitioning approach—specifically classification and regression trees[49 50] compared with multivariable regression (figure 1).

## Qualitative analysis methods

Audio-recordings will be transcribed verbatim and checked for accuracy and entered into NVivo V.10. Data will be independently coded and analysed by two members of the research team. We will adopt a similar approach to Pinnock,[51] undertaking a thematic and narrative analysis of interviews, exploring how perspectives evolve over time, with detailed attention to patient and family perspectives on experience of care in each setting and transitions, including potential interventions (and hence cost levers or other triggers) to influence changes in settings of care.

## Model development

The distribution of casemix criteria (cost predictors) in the participant population will be described based on phase of illness, functional status and problem severity, using parametric or non-parametric statistics, as appropriate. In order to reduce the number of cost predictors,

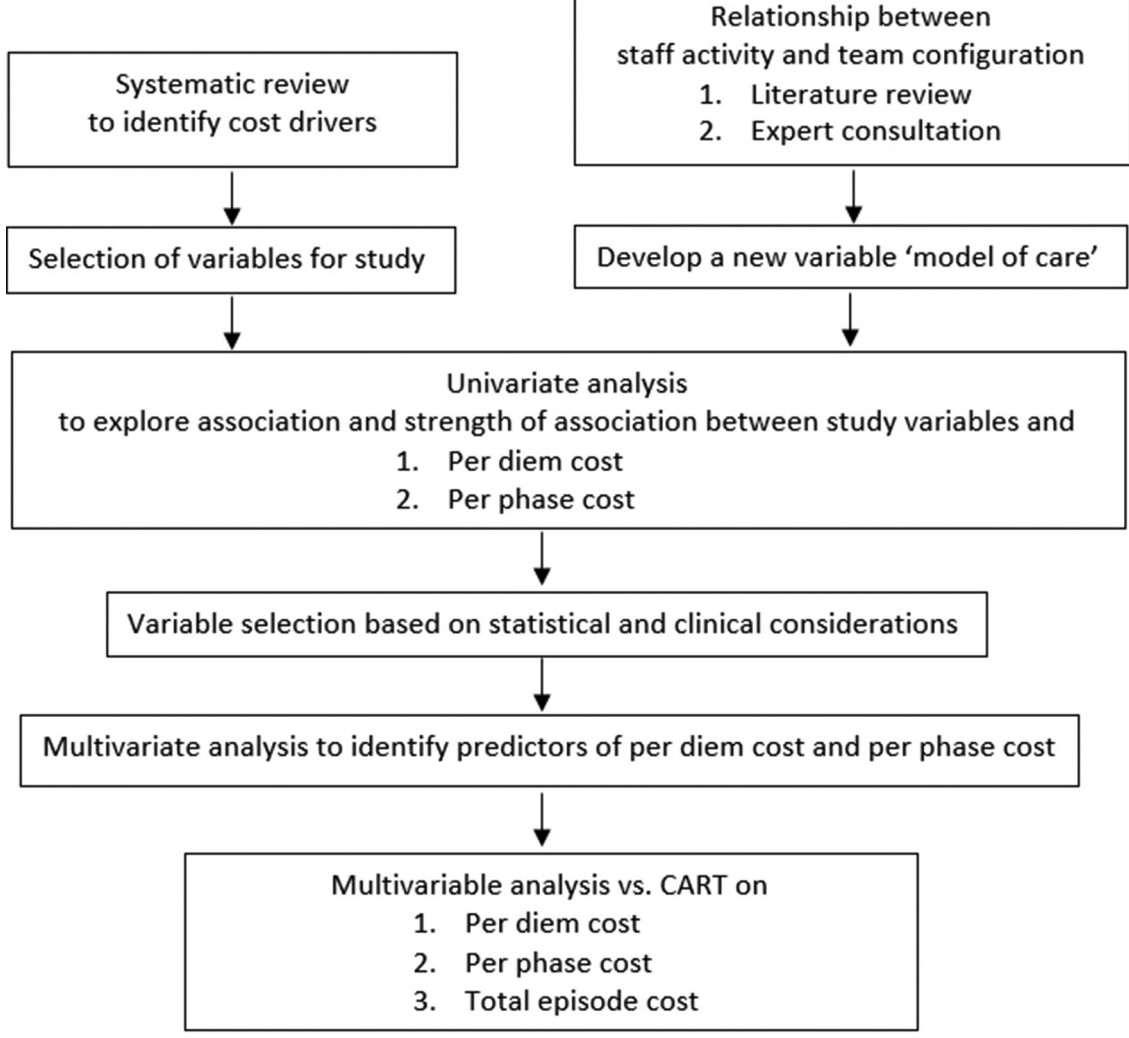

**Figure 1** Data analysis flowchart. CART, classification and regression trees.

we will initially use univariate analysis to explore the association and strength of association between casemix criteria and cost of the episode of care (quantifying cost by per diem cost, per phase cost and total cost of episode). We will use generalised linear model (multivariate regression model) to select variables that will then be applied in a hierarchical manner to form a branching classification in which each cost driver is incorporated only once.

We will select variables according to how much of the predictive error ($R^2$) each variable predicts and the related P value. After each bootstrap iteration, we will rank the variables according to these two criteria and remove the weakest (non-significant) predictor. Variables shown in tables 2 and 3 and the newly created categorical variable representing 'model of care' will be initially entered into the models and then non-significant predictors removed at each iteration using a Bootstrap subsampling strategy. We will create dummy variables to enable each level of phase of illness and model of care (categorical variables) to be assessed individually. Internal validation will be conducted by using such methods as bootstrapping or cross-validation to quantify any optimism in the predictive performance (calibration and discrimination) of the developed classification.

## Model specification

The casemix classification for each outcome across three settings will be presented including all regression coefficients and model intercept (figure 2). How to use the classification will be explained. Understanding how casemix variables predict cost of the episode of care enables a robust casemix classification to be developed. But in order for casemix adjustment to occur, there also needs to be a good understanding of how the casemix variables impact on clinical outcomes, as well as resource use. This casemix adjustment methodology is described in the recently published Department of Health document,[52] which recommends that significant casemix variables are identified, and that the size of their relationship with clinical outcomes is determined in advance.

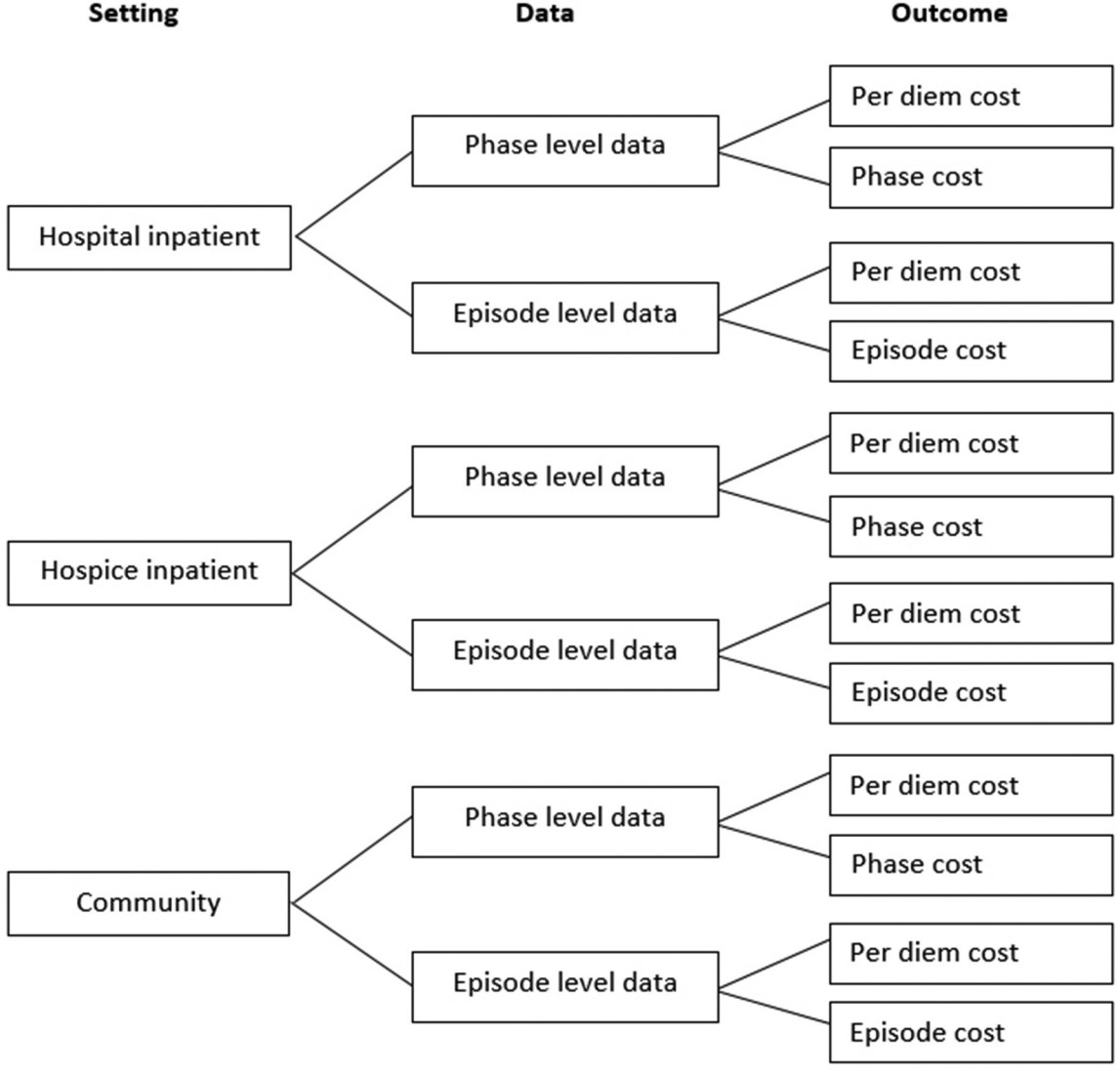

**Figure 2** Casemix classification specification.

Multilevel modelling will be undertaken to ensure that both of these steps will be delivered in this study by determining (1) which casemix criteria are most strongly associated with clinical outcomes, allowing for clustering at the level of 'participant' (where episodes occur in the same individual), at the level of 'model of care' (where care is received according to similar models), and at the level of 'site' (where care is delivered in a similar way within site), and (2) which casemix criteria are the strongest cost predictor (including per diem cost, per phase cost and total episode cost).

## Model performance

The performance of the casemix classification will be assessed in both the development and validation data sets. Classification measures including sensitivity, specificity, predictive values and net reclassification improvement will be reported and cut points selected a priori. We will primarily focus on discrimination for the classification development phase, and assess both discrimination and calibration for the validation phase.[53] Discrimination of the classification will be measured using the concordance statistic and CIs (c-statistic).[54] Calibration of the classification will be measured using calibration plot and Hosmer-Lemeshow goodness-of-fit test.[55]

## Development versus validation

Data will be collected from patients, caregivers and clinicians during the validation phase to compare patient-reported problem severity, that is, 'felt' need collected in validation phase only (as measured by the patient-reported IPOS) and professional-reported problem severity, that is, 'normative' need (as measured by the clinician-reported IPOS, phase of illness, AKPS), by reporting correlation across levels of complexity and across conditions. We will assess how professional-reported measures relate to patient-reported outcomes across different levels of complexity (ie, in relation to casemix classes), in order to determine optimal outcome measures and quality indicators.

We will compare data on resource use from two sources: (1) clinician-completed brief PRUS, and (2) patients and family-completed CSRI, to better understand how the PRUS maps to patients' receipt of resources, and evaluate which generic services are used across different providers, regions and the context of different models of specialist palliative care.

## Ethical considerations

The study protocol and documents (eg, the participant information sheet, consent and declaration form) have been approved by the NHS Health Research Authority London—Camberwell St Giles Research Ethics committee (15/LO/0887) for the development phase, and NHS Health Research Authority London—Bromley Research Ethics committee (16/LO/1021) for the validation phase.

All eligible participants are fully informed before consent is sought by the local research team or research nurses or project research team through the information sheets and verbal explanation on the aims and methods of the study and procedures that might be involved. Participants can withdraw at any time up to analysis of their data, without giving any reason. It is possible that participants may become distressed or raise issues during this study which raise concerns or warrant a change in their medical management, but we do not expect the questionnaires will themselves cause distress so much as uncover pre-existing distress which has not been acknowledged or recognised. Should this be the case, then our distress protocol will be followed. We anticipate distress will be infrequent, given the general nature of the questionnaires. It is likely that any distress will reflect advanced disease and experiences of care, and not the questionnaires themselves. All of the study team members have completed Good Clinical Practice training, and specific training on addressing distress in palliative care.

Our existing Patient/Public Advisory Group and extended Consumer Panel have been and will continue to be consulted throughout the study to ensure that the study is carried out in an ethical and respectful way, and has the highest possible relevance and benefit to patients and families. A Project Steering Committee meets once every six months to monitor recruitment, review the detailed progress of the study and make recommendations for overall direction and strategy.

## Dissemination

This study will lead to patient benefit through improved matching of resources to needs at individual patient-level and will better enable the NHS to deliver high-quality, patient-centred palliative care in last year of life. The results of the study will be published in peer-reviewed publications and will also be presented at national and international conferences.

**Author affiliations**
[1]Department of Palliative Care, Policy and Rehabilitation, Cicely Saunders Institute, King's College London, London, UK
[2]Department of Primary Care and Public Health Sciences, King's College London, London, UK
[3]University of Wollongong, Australian Health Services Research Institute, Centre for Health Service Development, Wollongong, Australia
[4]Wolfson Palliative Care Research Centre, Hull York Medical School, University of Hull, Hull, UK

**Acknowledgements** This article presents independent research C-CHANGE funded by the National Institute for Health Research (NIHR) under the Programme Grants for Applied Research scheme (RP-PG-1210-12015) and the NIHR CLAHRC funding scheme, through the Collaboration for Leadership in Applied Health Research and Care South London (NIHR CLAHRC South London) at King's College Hospital NHS Foundation Trust. The C-CHANGE research team acknowledges the support of the National Institute of Health Research Clinical Research Network (NIHR CRN).

**Contributors** PG drafted the study protocol with input from all authors. MD, JMD and AD provided statistical advice and support. AMF, SMO, CP, SP and PG contributed to the acquisition and interpretation of data for the work. IJH and

FEMM conceived the study and were in charge of overall direction and planning. KE provided experience from the Australian palliative care casemix development. FEMM is the chief investigator and designed the C-CHANGE research programme of which this is part. All authors provided critical feedback, helped revise the manuscript and are accountable for the accuracy and integrity of all aspects of the work.

**Funding** This work was supported by NIHR Programme Grants for Applied Research (grant number RP-PG-1210-12015).

**Disclaimer** The views expressed in this article are those of the author(s) and not necessarily those of the National Health Service, the National Institute for Health Research, or the Department of Health and Social Care.

**Competing interests** None declared.

**Patient consent** Detail has been removed from this case description/these case descriptions to ensure anonymity. The editors and reviewers have seen the detailed information available and are satisfied that the information backs up the case the authors are making.

**Ethics approval** The NHS Health Research Authority (HRA) Approval.

**Provenance and peer review** Not commissioned; externally peer reviewed.

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
