## [Reviewer comments · BMJ Open]

ARTICLE DETAILS

TITLE (PROVISIONAL)	Development and validation of a casemix classification to predict costs of specialist palliative care provision across inpatient hospice, hospital, and community settings in the UK: A study protocol
AUTHORS	Guo, Ping; Dzingina, Mendwas; Firth, Alice; Davies, Joanna; Douiri, Abdel; O'Brien, Suzanne; Pinto, Cathryn; Pask, Sophie; Higginson, Irene; Eagar, Kathy; Murtagh, Fliss

VERSION 1 – REVIEW

REVIEWER	Katja Grasic University of York, UK
REVIEW RETURNED	10-Nov-2017

GENERAL COMMENTS	I believe this is an important topic and there is a general lack of information on both, the activity levels as well as cost information for palliative care. While I enjoyed reading the paper, I believe there could be several improvements made in order for it to be clearer and more consistent. 1. The use of tense should be consistent; for example, authors jump between future and past tense when describing phase 1. On page 5 they say that "In phase 1, we will identify"... and then the tense is switched to past in the continuation. In this way, it's hard to follow what was already done and what not.2. Similar to the previous point, the authors could include some of the numbers that they already have. For example, for Phase 1 they say "Median duration of an episode of care is expected to be under 14 days in inpatient and at a median of 72 days in community settings". Assuming, this data has already been collected, actual numbers should be used instead.3. Authors mention that a similar case-mix classification exists in Australia, however, it cannot be directly transferred to UK settings. Authors then mention Australian system at various points but it's not really clear how the systems differ or to which extent did the Australian system serve as a template. There should be some more clarity about this.4. It's not clear how the participating sites were selected and how many there are (per setting). On page 6 it says that all patients in participating sites were selected for phase 1; in phase 2 it says that all patients from 14 sites were selected? Is this the same number as in phase 1?5. At several points authors mention unmet needs, for example, on page 7, they mention that in PRUS they will check whether the level of professional input met the patient/carers needs as well as details
--

	about any unmet need. It's unclear how unmet need is defined and how it is measured. 6. Staff will collect data on high-cost drugs, diagnostic tests and other information. How will this be measured? Using standardised OPCS codes or in any other way? 7. Authors say they will purposively select 20-25 patients for an interview, but it's unclear what characteristics will be taken into account for selection. 8. Several questionnaires are used in the process (for example Distress Thermometer, Pall-CSRI). If they are not too lengthy, they could be included in the appendix to give the reader clearer idea of the process. 9. On page 10 authors mention the Model of Care. How is this different from the information collected in Table 1 as this also measures staff involvement in care by different categories? 10. In table 2 it says they measure the phase of illness, but there is no mentioning of the duration of the phase. As they then calculate the cost per phase, isn't that dependent upon the duration? 11. In Figure 1 authors say they explore the association between per diem/phase cost and study variables. It would be useful to have a separate diagram for how the costs are calculated. Currently, there is some jumping around and cost calculation is mentioned at several different points.
--	---

REVIEWER	Dr. Christian Becker Helmholtz Zentrum München, Germany
REVIEW RETURNED	15-Nov-2017

GENERAL COMMENTS	Dear Authors I enjoyed reading your study protocol, which, in my opinion addresses important topic in health services provision. Below you find several points on which I believe you should provide additional information:  -Please provide a clear description of the study settings where data is collected (phase II) or has been collected (phase I) and provide a list of study sites or a reference to where this can be obtained. -Please describe study enrolment procedure more clearly. Did you include all patients who were being treated on a particular day or did you enroll people who were admitted starting this particular day? -Please provide a definition of "direct care costs" (that is, which cost components are included?) and describe the perspective of your cost calculation. For what reason did cost calculations in phase I and phase II have a different scope? -Was staff activity reported electronically or was this based on paper records? Were measures taken to promote data quality (for example, to ensure that the same definitions of activities were used across study sites)? -On page 11, you stated that you used the 10 events per variable rule in the validation phase. The only applications of this rule that I am aware of were cox regression and logistic regression models focusing on survival outcomes, rather than cost outcomes. Please provide a more detailed description on how you plan to apply this rule in applied in this study (for example, what would be considered an "event"). In addition, please check if the reference provided for the 10 events per variable rule is correct. -In Figure 1, for steps involving patient-level data, I would find it helpful to see whether the involved data was from phase I, phase II or from both phases.
---

	-Regarding model development, on page 13, you stated that you intended to use a generalized linear model to select variables. What are the criteria for variable selection? In addition, you stated that non-significant predictors would be removed at each iteration. How do you plan to handle categorical variables (such as "phase of care" or "model of care"), for which only some of the levels could be significant? -Who obtained informed consent and how? Please provide the informed consent forms that were used
--	--

REVIEWER	Kevin Whitford, MD, MS Mayo Clinic USA
REVIEW RETURNED	20-Nov-2017

GENERAL COMMENTS	Thank you for doing this important study. This study will provide further information to better understand case mix classification for palliative care and help determine the optimal model(s) of care.
---

VERSION 1 – AUTHOR RESPONSE

Thank you for these helpful editorial and review comments. We have thought about these very carefully. Our response and changes made are shown below.

Editorial Comment:

Re reviewer 1's comments: please do not present any results in this manuscript. Study protocols should present in detail the study's methodology and rationale. The study's results should be submitted separately as a research article.

Authors' response: Thank you so much for your confirmation. We won't present any results in this manuscript.

Editorial Request:

Please briefly elaborate on your dissemination plans in the Abstract >> 'Ethics and dissemination' section.

Authors' response: We have added the following to briefly elaborate on our dissemination plans in the abstract: The results are expected to be disseminated in 2018 through papers for publication in major palliative care journals; policy briefs for clinicians, commissioning leads, and policy makers; and lay summaries for patients and public.

Reviewers' Comments to Author:

Reviewer: 1

Reviewer Name: Katja Grasic

Institution and Country: University of York, UK Competing Interests: None

I believe this is an important topic and there is a general lack of information on both, the activity levels as well as cost information for palliative care. While I enjoyed reading the paper, I believe there could be several improvements made in order for it to be clearer and more consistent.

Authors' response: We are so glad to hear the recognition of the importance of this research topic and that you enjoyed reading our paper. We believe that your advice helped us to improve the overall clarity of our paper. Thank you so much.

1. The use of tense should be consistent; for example, authors jump between future and past tense when describing phase 1. On page 5 they say that "In phase I, we will identify"... and then the tense is switched to past in the continuation. In this way, it's hard to follow what was already done and what not.

Authors' response: Phase I data collection has been completed but data analysis is ongoing. Therefore, on page 5 we say that 'In phase I, we will identify variables that predict costs to be included in casemix classification.....' and later when we describe the data collection, we say that 'Phase I was conducted between 31st July 2015 and 30th September 2016 and follow-up ended 30th November 2016. Data were collected from clinicians through surveys.....' Phase II data collection is being undertaken, and we use future tense consistently. Would the editor advise whether we should either use future tense for both phases or leave it as it is currently?

2. Similar to the previous point, the authors could include some of the numbers that they already have. For example, for Phase 1 they say "Median duration of an episode of care is expected to be under 14 days in inpatient and at a median of 72 days in community settings". Assuming, this data has already been collected, actual numbers should be used instead.

Authors' response: Following the editor's comments, no changes has been made and we have left out any results in this manuscript.

3. Authors mention that a similar case-mix classification exists in Australia, however, it cannot be directly transferred to UK settings. Authors then mention Australian system at various points but it's not really clear how the systems differ or to which extent did the Australian system serve as a template. There should be some more clarity about this.

Authors' response: Even though we had already mentioned on page 4 that 'due to variations in outcome measures collected and palliative care provided between countries, it is unclear whether any existing palliative care classification can be easily applied to the UK to address unmet needs and resolve the inequity', we added the following sentences to clarify this: The Australian casemix classification consists of classes defined by five criteria including phase of illness, problem severity, functional status and dependency, age, model of care, as most strongly predictive of resource use.¹⁶ Full class definition and categorisation is available at <http://www.pcoc.org.au/>. In addition, in the Statistical analysis methods section on page 12, we had also mentioned that 'In the Australian classification, five casemix criteria (phase of illness, problem severity, functional status and dependency, age, model of care) were found to be most strongly predictive of resource use. If our analyses demonstrate the same five criteria as most predictive of resource use, the Australian classification or a refinement of it will be adopted. If there is disparity, then the data on individual casemix criteria will be used to develop a new casemix classification, using a recursive partitioning approach – specifically classification and regression trees (CART) 48, 49 compared to multivariable regression (Figure 1).'

4. It's not clear how the participating sites were selected and how many there are (per setting). On page 6 it says that all patients in participating sites were selected for phase 1; in phase 2 it says that all patients from 14 sites were selected? Is this the same number as in phase 1?

Authors' response: In the participants section on page 6 and 7, we have explained how the participating sites were selected and how many there are, in order to clarify this.

Phase I – Development phase

All adults (≥18 years) receiving specialist palliative care newly admitted in 10 participating sites during the study period (2 sites providing hospital advisory services, 1 providing community-based service only, 6 providing hospice inpatient and community-based services, 1 providing hospital advisory and community-based services) were included, regardless of primary diagnosis. We selected these sites with the aim of ensuring a representative sample in terms of population demographics (age distribution, ethnicity, socio-economic status, and rural/urban composition).

Phase II- Validation phase

All adults (≥18 years) receiving specialist palliative care newly admitted from 14 sites during the study period (4 extra sites to increase recruitment) will be eligible, regardless of primary diagnosis.

5. At several points authors mention unmet needs, for example, on page 7, they mention that in PRUS they will check whether the level of professional input met the patient/carers needs as well as details about any unmet need. It's unclear how unmet need is defined and how it is measured.

Authors' response: On page 8, we added the following sentence to explain how unmet need is measured: 'When completing PRUS, the member of staff is asked to estimate if any patients' or families' unmet needs existed, and if so, what needs were unmet.'

6. Staff will collect data on high-cost drugs, diagnostic tests and other information. How will this be measured? Using standardised OPCS codes or in any other way?

Authors' response: On page 8, we have added: Within PRUS, the member of staff is asked to indicate whether or not equipment, high cost oral/transdermal medications, injectable medications/interventions, and medical imaging or tests were received by the participant, from a list derived from prior work to identify these options.

7. Authors say they will purposively select 20-25 patients for an interview, but it's unclear what characteristics will be taken into account for selection.

Authors' response: On page 7, we have added: To capture variation in age, gender, diagnosis, and geographical location, we will purposively select 20-25 patients and family caregivers with at least two transitions of care for face-to-face interviews.

8. Several questionnaires are used in the process (for example Distress Thermometer, Pall-CSRI). If they are not too lengthy, they could be included in the appendix to give the reader clearer idea of the process.

Authors' response: To clarify this, on page 9, we have added: The measures are available direct from: Phase of Illness (www.pcoc.org.au), AKPS,³⁹ Modified Barthel Index,⁴⁰ and IPOS (www.pos-pal.org) respectively.

9. On page 10 authors mention the Model of Care. How is this different from the information collected in Table 1 as this also measures staff involvement in care by different categories?

Authors' response: Table 1 is staff activity recorded by using the time (minutes) spent on face-to-face and phone contacts, and patient-level administrative time per shift. However, the Model of Care is the

configurations of staff, interventions, and other elements which together make up the service, as reported in the Model of Care section on page 11.

10. In table 2 it says they measure the phase of illness, but there is no mentioning of the duration of the phase. As they then calculate the cost per phase, isn't that dependent upon the duration?

Authors' response: For purpose of casemix, Phase of Illness at start of episode is used. We have clarified this in the Table 2. The cost per day (per diem cost) would be dependent upon the duration of the episode, not phase.

11. In Figure 1 authors say they explore the association between per diem/phase cost and study variables. It would be useful to have a separate diagram for how the costs are calculated. Currently, there is some jumping around and cost calculation is mentioned at several different points.

Authors' response: A standard costing methodology adopted from the current NHS costing principles³⁸ will be used. It has been described on page 9: The costs of each resource item will be calculated by combining the resource use data with appropriate unit cost obtained from recognised sources including the annual compendium produced by the Personal Social Services Research Unit (PSSRU),³² prices on the NHS supply chain website <http://www.supplychain.nhs.uk>, British national formulary,³³ and NHS reference costs.³⁴ The focus will be on staff time - the main resource in palliative care documented in a variety of settings and countries.³⁵⁻³⁷ We do not think a separate diagram will add more, beyond this explanation.

Reviewer: 2

Reviewer Name: Dr. Christian Becker

Institution and Country: Helmholtz Zentrum München, Germany Competing Interests: None declared

Dear Authors

I enjoyed reading your study protocol, which, in my opinion addresses important topic in health services provision. Below you find several points on which I believe you should provide additional information:

1. Please provide a clear description of the study settings where data is collected (phase II) or has been collected (phase I) and provide a list of study sites or a reference to where this can be obtained.

Authors' response: A list of study sites will shortly be available on our C-CHANGE project webpage via: <https://www.kcl.ac.uk/nursing/departments/cicelysaunders/research/studies/c-change/c-change.aspx>. We will ensure this is available before publication. On page 7, we have added: For both Phase I and II, a list of study sites can be obtained via: <https://www.kcl.ac.uk/nursing/departments/cicelysaunders/research/studies/c-change/c-change.aspx>.

2. Please describe study enrolment procedure more clearly. Did you include all patients who were being treated on a particular day or did you enroll people who were admitted starting this particular day?

Authors' response: We enrolled new patients who were admitted or entered to a service during the study period. To clarify this, in the participants section phase I, we have added:

Phase I – Development phase

All adults (≥18 years) receiving specialist palliative care newly admitted in 10 participating sites during the study period (2 sites providing hospital advisory services, 1 providing community-based service

only, 6 providing hospice inpatient and community-based services, 1 providing hospital advisory and community-based services) were included, regardless of primary diagnosis.

Phase II – Validation phase

All adults (≥ 18 years) receiving specialist palliative care newly admitted from 14 sites during the study period (4 extra sites to increase recruitment) will be eligible, regardless of primary diagnosis.

3. Please provide a definition of “direct care costs” (that is, which cost components are included?) and describe the perspective of your cost calculation. For what reason did cost calculations in phase I and phase II have a different scope?

The following has been added on page 8 to define direct care costs and describe our cost perspective for both phases: ‘Direct care cost refers to all costs due to resource use that are completely attributable to the use of a health care intervention or illness, which can be split into direct medical costs (the cost of a defined intervention and all follow-up costs for other medication and health care interventions in ambulatory, inpatient, nursing care, home, or other relevant settings) and direct non-medical costs (e.g., transportation costs and additional paid caregiver time).³² In the validation phase, we will use the same health services costing perspective as in the development phase, but informal care costs are also considered to be in-scope and will be analysed separately.’

4. Was staff activity reported electronically or was this based on paper records? Were measures taken to promote data quality (for example, to ensure that the same definitions of activities were used across study sites)?

Authors’ response: Staff activity was reported based on paper records as described in the manuscript on page 7: At every contact, nurses, doctors, and allied health professionals will record the time spent on face-to-face and phone contacts, and patient-level administrative time per shift using the staff activity matrix paper version (Table 1). Then the following has been added: Staff training and site feedback of activity data were conducted regularly to improve and optimise data quality.

5. On page 11, you stated that you used the 10 events per variable rule in the validation phase. The only applications of this rule that I am aware of were cox regression and logistic regression models focusing on survival outcomes, rather than cost outcomes. Please provide a more detailed description on how you plan to apply this rule in applied in this study (for example, what would be considered an “event”). In addition, please check if the reference provided for the 10 events per variable rule is correct.

Authors’ response: We accept the limitations of the “10 events per variable” rule for the validation phase, and recognise that we are not handling ‘events’ in this analysis. Therefore we have reverted to the same rule as we adopted in Phase I and revised the paragraph:

In the validation phase, again a minimum of $50 + 8 \times m$ (where m is the number of predictors) is required to test the hypothesis that the population multiple correlation equals zero with a power of 80%, $\alpha = 5\%$ and a medium effect size for the regression analysis ($R^2 = 0.13$).⁴⁶ In Phase II, the number of predictors reflects the casemix variables, and needs to be clinically relevant⁴⁷ to ensure a meaningful case-mix classification. We expect the final casemix classification to contain 5-8 casemix variables, based on the Australian experience.¹³ We have estimated that data from 114 ($50 + (8 \times 8)$) episodes of care per setting (hospice, hospital, community) are needed. Assuming an average of three episodes per participant¹⁵ in this longitudinal phase, this represents at least 38 participants per setting. Allowing for 50% attrition, we estimate needing a total of 228 participants, which will provide about 684 patient episodes. The sample size will be re-calculated and inflated using the design effect based

on intraclass correlation which will be estimated through interim analyses. A total of 300 participants across settings is needed, including extra 25% participants allowing for intraclass correlation.

6. In Figure 1, for steps involving patient-level data, I would find it helpful to see whether the involved data was from phase I, phase II or from both phases.

Authors' response: The involved the data was from both phases.

7. Regarding model development, on page 13, you stated that you intended to use a generalized linear model to select variables. What are the criteria for variable selection? In addition, you stated that non-significant predictors would be removed at each iteration. How do you plan to handle categorical variables (such as "phase of care" or "model of care"), for which only some of the levels could be significant?

Authors' response: On page 14, we have added detail about the criteria for variable selection: "We will select variables according to how much of the predictive error (R-square) each variable predicts and the related p-value. After each bootstrap iteration we will rank the variables according to these two criteria and remove the weakest (non-significant) predictor".

On page 14, we have added detail of how we will handle categorical variables: "We will create dummy variables to enable each level of Phase of Illness and Model of care (categorical variables) to be assessed individually."

8. Who obtained informed consent and how? Please provide the informed consent forms that were used.

Authors' response: Informed consent is obtained by local research team or research nurses or project research team. This information is now added in the Ethical considerations section on page 16. We used several forms including phase I consent form, phase I declaration form (for gaining assent), phase II quantitative consent form, phase II quantitative declaration form, phase II interview consent form. We have explained how we obtained consent/assent in the manuscript for both phases, therefore we do not think that including all forms that were used in the manuscript will add more, beyond this explanation. However, we are happy to share our forms upon request from the readers. Thank you very much.

Reviewer: 3

Reviewer Name: Kevin Whitford, MD, MS

Institution and Country: Mayo Clinic, USA

Competing Interests: None declared

Thank you for doing this important study. This study will provide further information to better understand case mix classification for palliative care and help determine the optimal model(s) of care.

Authors' response: Thank you so much for your recognition of the importance of this study, which will provide information to better understand case mix classification for palliative care and help to determine the optimal models of care.

VERSION 2 – REVIEW

REVIEWER	Christian Becker Helmholtz Zentrum München, Germany
REVIEW RETURNED	16-Jan-2018

GENERAL COMMENTS	Thank you for addressing the points I raised in my review. I believe through your answers the protocol has gained in clarity and detail. My comment on the need to publish your informed consent forms was based on the SPIRIT statement. Since your reporting is based on the TRIPOD checklist, which does not ask for consent forms, not including these in this study protocol appears also justified.
---

REVIEWER	Katja Grasic University of York, UK
REVIEW RETURNED	29-Jan-2018

GENERAL COMMENTS	Thank you for the revised version of the paper. I believe all points were well addressed. Some further suggestions below: 1. It would be clearer for the reader if the sentence would read "In the analysis part of Phase 1 we will...". I think it's ok to use future tense for activities that were not yet completed, it just needs to be clear what was done and what not. 4. Can you add a sentence about how the additional sites will be identified? 5. I am still a bit unclear by what is measured by an unmet need and how it is measured. Is there any standardised form/process by which the staff identifies unmet need?
--

VERSION 2 – AUTHOR RESPONSE

Reviewer: 1

Reviewer Name: Katja Grasic

Institution and Country: University of York, UK Competing Interests: None declared.

Thank you for the revised version of the paper. I believe all points were well addressed. Some further suggestions below:

1. It would be clearer for the reader if the sentence would read "In the analysis part of Phase 1 we will...". I think it's ok to use future tense for activities that were not yet completed, it just needs to be clear what was done and what not.

Authors' response: Thank you for your suggestion. We have revised this sentence as suggested.

4. Can you add a sentence about how the additional sites will be identified?

Authors' response: We have added a sentence about this: All adults (≥18 years) receiving specialist palliative care newly admitted from 14 sites during the study period (participating sites in Phase I, but extending to four additional sites to increase recruitment) will be eligible, regardless of primary diagnosis. Additional sites were accepted if they could contribute to maintaining or extending representativeness in terms of population demographics (age distribution, ethnicity, socio-economic status, and rural/urban composition).

5. I am still a bit unclear by what is measured by an unmet need and how it is measured. Is there any standardised form/process by which the staff identifies unmet need?

Authors' response: We use Palliative care Resource Score (PRUS) to capture unmet needs. To improve clarity, on page 8, we introduce PRUS as below: PRUS is a questionnaire specifically designed to capture palliative care resource use in a standardised way. Then we further explain how unmet needs are captured by Yes/No question and free text space across several main areas of care provision: Any unmet needs which provide insight into any gaps between needs and provision, as identified by a member of staff.²⁷⁻²⁹ When completing PRUS, the member of staff is asked to estimate retrospectively if there were any unmet needs (Yes/No) in the care provided by health care assistants, registered nurses, palliative medicine doctors, and allied health care professionals. This member of staff is also asked whether there were any unmet needs in out of hours care, and whether the equipment provided met the patient's needs. If there were unmet needs identified in any of these areas above, a free text field is provided to specify these unmet needs.

Reviewer: 2

Reviewer Name: Christian Becker

Institution and Country: Helmholtz Zentrum München, Germany Competing Interests: None declared

Thank you for addressing the points I raised in my review. I believe through your answers the protocol has gained in clarity and detail. My comment on the need to publish your informed consent forms was based on the SPIRIT statement. Since your reporting is based on the TRIPOD checklist, which does not ask for consent forms, not including these in this study protocol appears also justified.

Authors' response: Thank you so much for your confirmation. We won't include these consent forms in this study protocol.